# Effects of Energy Dissipation Pier Arrangements on the Hydraulic Characteristics of Segmented Pier-Type Step Energy Dissipator Structures

**Ziwei Feng, Yongye Li \*, Yu Tian and Qian Li**

College of Water Resource Science and Engineering, Taiyuan University of Technology, Taiyuan 030024, China
\*  Correspondence: liyongye@tyut.edu.cn; Tel.: +86-139-3423-9832

**Abstract:** To make more efficient use of limited space, improve the energy dissipation effect of the step dissipator, and mitigate the effect of cavitation, we propose a segmented pier-type step dissipator structure and used a numerical simulation to study the hydraulic effects of two different arrangements of piers: a double-row arrangement and a staggered arrangement. We've drawn the following conclusions from our study: the segmented pier-type structure produces a large water jump at the location of the energy dissipation pier. This involves a large amount of air, promotes air-doping of the water flow in the whole section, and reduces the length of the non-air-doping zone. The staggered pier arrangement produces a better air-doping effect at the water jump and a higher air-doping concentration along the water course. The staggered arrangement also produces a better cavitation mitigation effect and is better able to stabilise the water flow; the flow velocity at the outlet is lower, so the energy dissipation effect is better. A larger positive pressure area forms at the headwater and upstream areas of the energy dissipation pier; a larger negative pressure forms at the top and backwater of the energy areas. The staggered arrangement produces a larger negative pressure; however, under various flow conditions, the difference in the energy dissipation rate between the two forms of pier arrangements is not significant. We obtained a peak energy dissipation rate of 90.04%, which represents an improved energy dissipation effect compared with the control. The step energy dissipator described here is conducive to stabilising the outlet flow, reducing cavitation damage, and improving energy dissipation. These findings provide a valuable reference for the future design of sectional pier-type step energy dissipator structures.

**Keywords:** step energy dissipator; trapezoidal energy dissipation pier; aerating effect; hydraulic characteristics; energy dissipation characteristics

## 1. Introduction

The concept of step energy dissipation has existed for more than 2500 years in water conservation projects. Since the second half of the 20th century, compacted concrete technology has developed rapidly. Its use in modern dam-construction projects has led to the application of step energy dissipation. In order to investigate the laws of energy dissipation, achieve better dissipation rates, and apply more scientific methods, scholars have conducted extensive research on the hydraulic characteristics of step spillway energy dissipation structures.

Li et al. [1] addressed the flow pattern problem through experimental observation and analysis. They divided the step water flow into a sliding region and a vortex region. Guo [2] provided a detailed description of the three flow patterns of the step spillway: sliding flow, transitional flow, and falling flow. Chakib Bentalha [3] and Tian [4], through experimental research, obtained the critical formula for the classification of the flow pattern under different slopes. Concerning pressure characteristics, Tian et al. [5] found that on the initial step surface, the hourly average pressure acting on the horizontal surface was positive, and the maximum pressure gradually shifted from the middle of the step to the

convex corner of the step as the flow increased; the negative pressure, which was recorded in the upper range of the vertical step surface when the flow decreased, became positive after entering the transition flow. Sánchez-Juny, M et al. [6] used model tests to investigate negative pressure on the horizontal surface of the step spillway and found that negative pressure on the horizontal surface was mainly located at 0.2 to 0.5 times the length of the horizontal step from the concave angle of the step; the higher the single-width flow, the higher the maximum value of pressure on the horizontal step, and the smaller the minimum value. Xu et al. [7] studied gas-doping along the course of the water flow and found that this process could be divided into three parts: a non-gas-doping zone, a gas-doping development zone, and a fully developed gas-doping zone, in which the non-gas-doping zone and the gas-doping development zone of the step surface are susceptible to cavitation damage. Zou [8] gave a detailed introduction to the mechanisms of gas-doping and gas-doping corrosion reduction and found through experimental research that the gas-doping concentration of the water flow rises as the step size increases. Wang et al. [9] determined the location of the initial dopant point on the step face and found that it shifted downwards as the single-width flow increased. The experimental findings of C.A. Gonzalez et al. [10] introduced the equation of the initial dopant point and the distribution law of dopant concentration in the section. Concerning the energy dissipation rate, Wen et al. [11] found, through model experiments on different step heights with fixed slope ratios, that the specific energy of water flow appears to increase first and then stabilise along the course. They also obtained an empirical formula for calculating residual energy. Zhang et al. [12] concluded through model tests that the step-doping and energy dissipation achieve optimal levels when the slope angle is between 30° and 45°. Tension et al. [13] conducted hydraulic model tests on step spillways in combination with real-world construction projects and concluded that step spillway dams can be widely used in low-head and low-flow water conservation projects. Jia [14] applied numerical simulations to compare and analyse several new step spillways and found that the conventional step type produced the highest average doping concentration, while the energy dissipation rate of the Kan-type step spillway was about 10% higher than the conventional design. Ma et al. [15] used relative hydraulic parameters and found that the relative energy dissipation rate along the course of the step spillway exhibited a linear growth trend, demonstrating a good linear correlation; Ma S.H.et al. [16] found that the relative flow velocity showed a good linear relationship with the number of sections and this relationship was not affected by critical water depth or slope. At present, most studies on step energy dissipation focus on adjusting such parameters as slope and step size. Li et al. [17–19] studied different step shapes and found that trapezoidal, concave-angle steps cause the initial air-mixing point to move forward. Gao et al. [20] proposed an improved coefficient method by combining the method of Changsang with model test results and thus determined a means of calculating the water depth of a reduced spillway. Amir Ghaderi et al. [21] proposed a trapezoidal, labyrinth-type step spillway, which they found was better able to interfere with the flow line and produced an obvious improvement in energy dissipation efficiency. Finally, Zhang et al. [22] studied the effect of front-doping on the energy dissipation of the step spillway by setting a front-doping can. Their results showed that front-doping can produce a certain effect on the doping effect of the step spillway.

In summary, researchers have conducted a large number of studies on step spillways. We note, especially, the finding that there is negative pressure near the top of the vertical surface of the step, which is known to cause cavitation damage. We also note that increasing the gas-doping concentration of the step water can reduce the cavitation hazard. Therefore, in order to improve the doping effect of the step water flow, increase the energy dissipation rate, and provide a certain reference basis for the design of the relevant structures, the segmented pier-type step energy dissipation structure is proposed, and the numerical simulation method is used to study the influence of the arrangement of the energy dissipation pier on the hydraulic characteristics of the segmented pier-type step energy dissipation work and explore the best arrangement of the energy dissipation pier.

## 2. Materials and Methods

### 2.1. Construction of Numerical Models

#### 2.1.1. Turbulence Model

For this study, we used Fluent software for the numerical simulation. Compared with the standard $k$-$\varepsilon$ model, the RNG $k$-$\varepsilon$ model adds an additional term to the $\varepsilon$ equation, which makes it more accurate when simulating flow fields with large velocity gradients [23–25]. In addition, the reforming group statistical technique makes the model more powerful for calculation purposes. We, therefore, used the RNG $k$-$\varepsilon$ turbulence model for our calculations, with the control equation as follows:

Continuity equations:

$$\frac{\partial \rho}{\partial t} + \frac{\partial (\rho u_i)}{\partial x_i} = 0 \tag{1}$$

Momentum equation:

$$\frac{\partial (\rho u_i)}{\partial t} + \frac{\partial (\rho u_i u_j)}{\partial x_i} = \frac{\partial \rho}{\partial x_i} + \frac{\partial (u \frac{\partial u_i}{\partial x_i} - \rho \overline{u_i' u_j'})}{\partial x_i} \tag{2}$$

$K$ equation:

$$\frac{\partial (\rho \kappa)}{\partial t} + \frac{\partial (\rho \kappa u_i)}{\partial x_i} = \frac{\partial}{\partial x_j} (\alpha_\kappa \mu_{eff} \frac{\partial \kappa}{\partial x_j}) + G_\kappa + \rho \varepsilon \tag{3}$$

$\varepsilon$ equation:

$$\frac{\partial (\rho \varepsilon)}{\partial t} + \frac{\partial (\rho \varepsilon u_i)}{\partial x_i} = \frac{\partial}{\partial x_j} (\alpha_\varepsilon \mu_{eff} \frac{\partial \varepsilon}{\partial x_j}) + \frac{C_{1\varepsilon}^*}{\kappa} G_\kappa - C_{2\varepsilon} \rho \frac{\varepsilon^2}{\kappa} \tag{4}$$

$$\mu_{eff} = \mu + \mu_t \tag{5}$$

$$\mu_t = \rho C_t \frac{\kappa^2}{\varepsilon} \tag{6}$$

$$C_{1\varepsilon}^* = C_{1\varepsilon} - \frac{\eta(1 - \eta/\eta_0)}{1 + \beta \eta^3} \tag{7}$$

$$\eta = (2E_{ij}E_{ij})^{\frac{1}{2}} \frac{\kappa}{\varepsilon} \tag{8}$$

$$E_{ij} = \frac{1}{2}(\frac{\partial u_i}{\partial x_j} + \frac{\partial u_j}{\partial x_i}) \tag{9}$$

where $\rho$ and $\mu$ denote the density and molecular viscosity coefficients derived from the weighted average of the volume fractions, respectively; $G_k$ denotes the turbulent kinetic energy generation term due to the mean flow velocity gradient; $\mu_t$ denotes the turbulent viscosity coefficient; $\mu_{eff}$ is the effective viscosity; $C_{1\varepsilon}$ and $C_{2\varepsilon}$ are the turbulent model coefficients, taking the values of 1.44 and 1.92, respectively; $\alpha_\kappa$ and $\alpha_\varepsilon$ denote the inverse of the Planck number of $k$ and $\varepsilon$, taking the values of 1.0 and 1.3, respectively; $C_\mu$ is the empirical coefficient, which takes the value of 0.0845; $\eta_0$ and $\beta$ are the model constants, which take the values of 4.38 and 0.015, respectively. See Table 1 for details.

**Table 1.** Study parameters and their range of variations.

| Parameters | Range of Variations |
|---|---|
| Q | 20 m$^3$/h, 30 m$^3$/h, 40 m$^3$/h, 50 m$^3$/h, 60 m$^3$/h |
| $C_{1\varepsilon}$ and $C_{2\varepsilon}$ | 1.44 and 1.92 |
| $\alpha_\kappa$ and $\alpha_\varepsilon$ | 1.0 and 1.3 |
| $C_\mu$ | 0.0845 |
| $\eta_0$ and $\beta$ | 4.38 and 0.015 |

### 2.1.2. Model Building

The numerical model consists of three parts: the inlet area, the stage, and the outlet, as shown in Figure 1. The research object is an energy dissipation structure involving 3 combined sections of steps and flat surfaces. The total length is 5.41 m, and the total elevation of the steps is 1.08 m. Each step has a width of 20 cm, a length of 18 cm, and a height of 6 cm. The trapezoidal energy dissipation pier size is 2 cm × 2 cm × 2 cm; the trapezoidal part size is 2 cm × 1 cm.

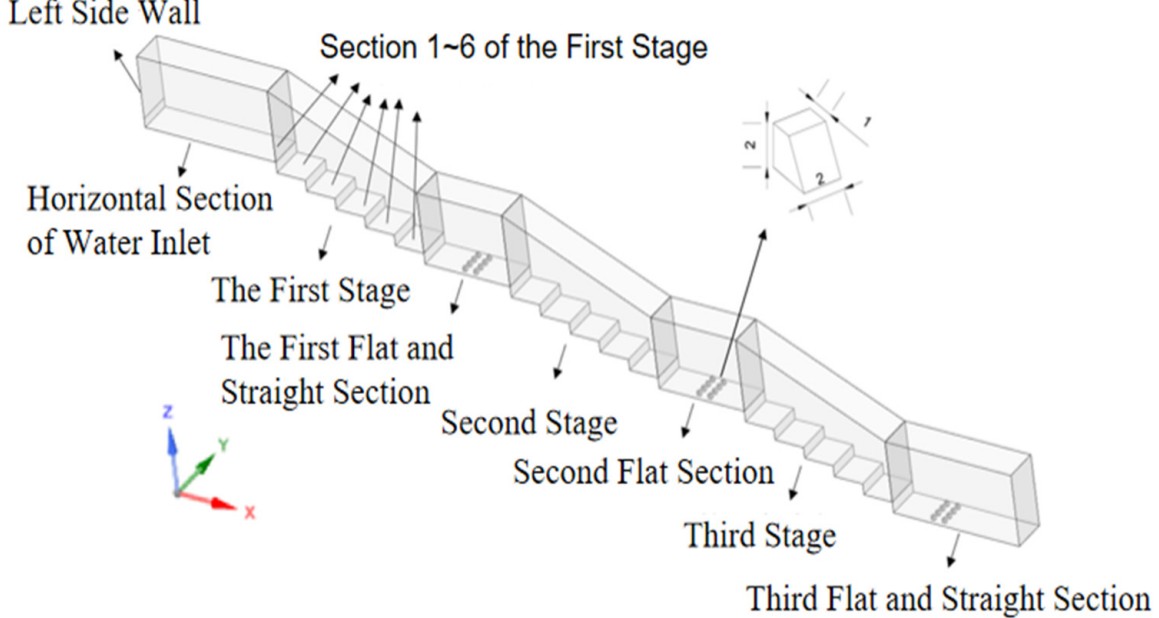

**Figure 1.** Schematic diagram of the numerical model.

The trapezoidal energy dissipation piers are arranged in 2 rows, themselves comprising 8 rows, at a position of 22.5 cm in the X-axis direction of each straight section. We used three different pier arrangements for our study purposes: (1) a double-row arrangement (two rows of dissipative piers in alignment); (2) a staggered arrangement (a staggered placement of two rows of piers); and (3) a control/traditional arrangement (no piers). The longitudinal position is 2.4 cm from the side wall, and the longitudinal spacing of the piers is also 2.4 cm. In the staggered arrangement, the first row of the piers is shifted by 1 cm as a whole, and the second row is moved to the right by 1 cm, as shown in Figure 2. Six different flow values were selected, as shown in Table 1.

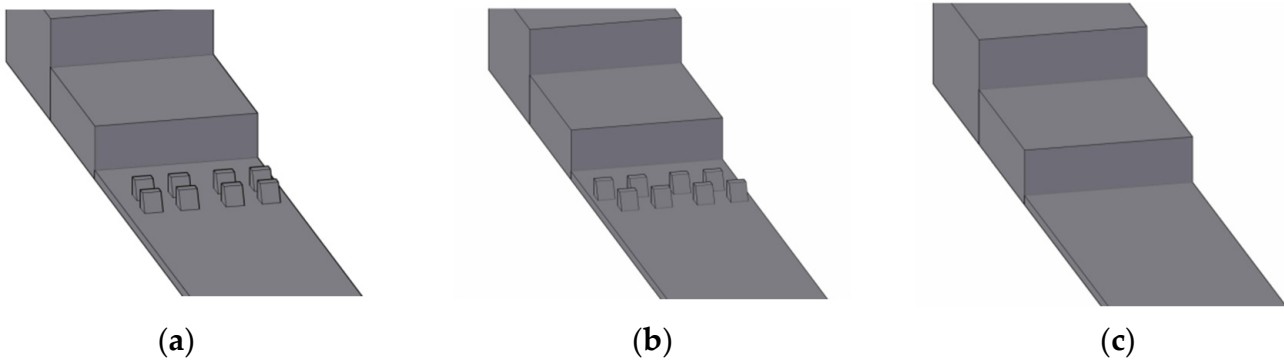

(**a**)　　　　　　　　　　　　　　(**b**)　　　　　　　　　　　　　　(**c**)

**Figure 2.** Schematic diagram of the arrangement of energy dissipation pier. (**a**) Double-row arrangement, (**b**) staggered arrangement, and (**c**) traditional steps.

### 2.1.3. Mesh Division and Boundary Setting

For meshing purposes, we used ANSYS native software, specifying a hexahedral structured mesh with a mesh size of 0.006 m. We used the PISO algorithm because it requires fewer iterations and converges more readily compared to other algorithms, thus saving time in the calculation of the transients. When determining boundary conditions, because the step energy dissipator water flow belongs to the gas–liquid two-phase flow, we divided the inlet boundary into two parts, according to the data measured by the physical test, as follows: the water inlet below adopts the flow inlet, and the air inlet adopts the pressure inlet. We set the top of the step to atmospheric pressure; the outlet boundary is the pressure outlet, and the side wall boundary is the solid no-slip boundary.

### 2.1.4. Mesh Sensitivity Analysis

To balance the need for computational accuracy with the requirements of time and resource cost, we carried out a mesh-independence test to determine the mesh size. As a reference, we selected the average water-depth variation at the convex corner of the third step at a flow rate of $Q = 40$ m$^3$/h. We then carried out a simulation for six different mesh sizes of 0.004 m, 0.005 m, 0.006 m, 0.007 m, 0.008 m, and 0.009 m. The results are shown in Figure 3.

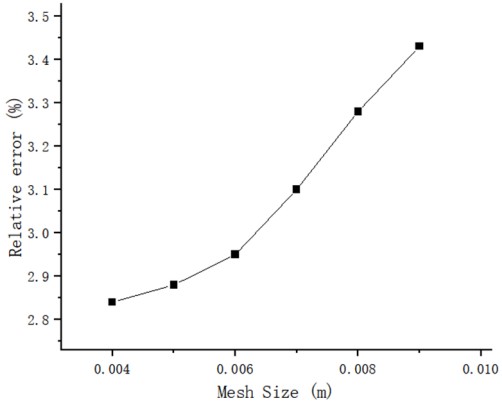

**Figure 3.** Mesh sensitivity analysis.

It can be seen that the water depth at the convex angle of the third step decreases as the mesh size decreases. In addition, the discrepancy between the simulation results for the mesh sizes of 0.006 m and 0.005 m is only 0.3%. We, therefore, conclude that mesh size is not the main factor affecting the simulation results at this time when there is a strong correspondence between the simulation results and the physical test results. Because a mesh size of 0.006 m was suitable for the study purposes, this was the size selected for our simulation, with a total number of meshes of approximately 1.1 million.

### 2.2. Experimental Validation of Numerical Model

Figure 4 shows the test system used in the physical model test, which includes valves, centrifugal pumps, underground reservoirs, electromagnetic flow meters, water stabilization tanks, and drainage tanks. The water is pumped from the underground reservoir to the stable-water tank through the centrifugal pump. The flow rate is controlled by the valve and is measured by the electromagnetic flowmeter. After the water attains a stable flow state in the stable-water tank, it is then freely discharged into the step test section, where the water depth and flow rate are measured using the measuring needle and the rotary slurry flow meter. The water is finally discharged into the underground reservoir through the drainage pool, and the circulation system cycle is completed.

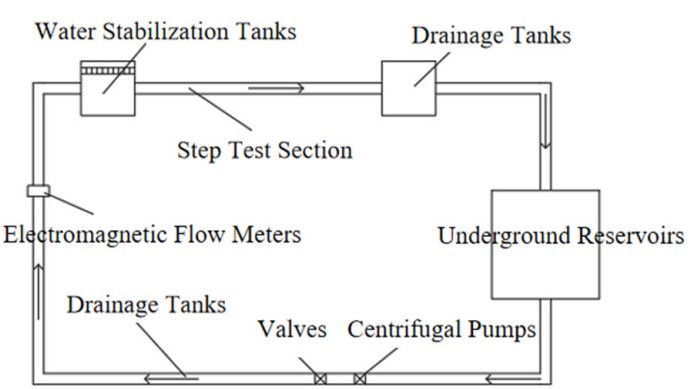
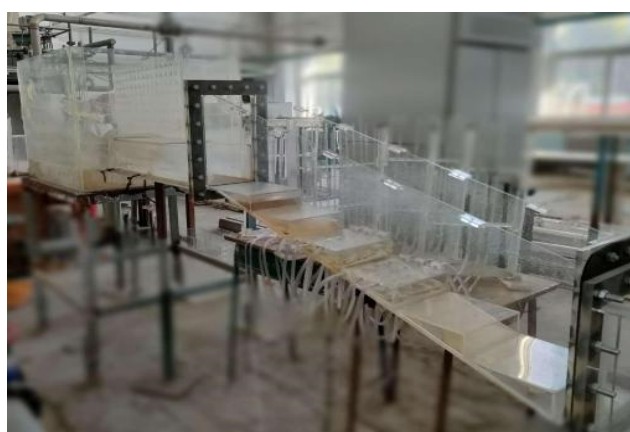

**Figure 4.** Schematic diagram of the test system.

The test section consists of the first step + the straight section and the first 20 cm of the stage. Figure 5 shows a schematic diagram of the measurement point layout. Under the working condition of the flow rate, Q = 30 m³/h, using a numerical simulation, we obtained values for the water depth, flow rate, and pressure along the distance of the double-row arrangement and compared these with the measured values of the corresponding physical tests. The pressure value was measured at the level of the fourth and fifth steps. Test results are presented in Figure 6 and show good agreement of the simulated and measured values for water depth, flow velocity, and pressure along the course, with maximum relative discrepancies of 7.09%, 6.54%, and 6.46%, respectively, under the double-row arrangement. These results confirm the feasibility of the simulation method.

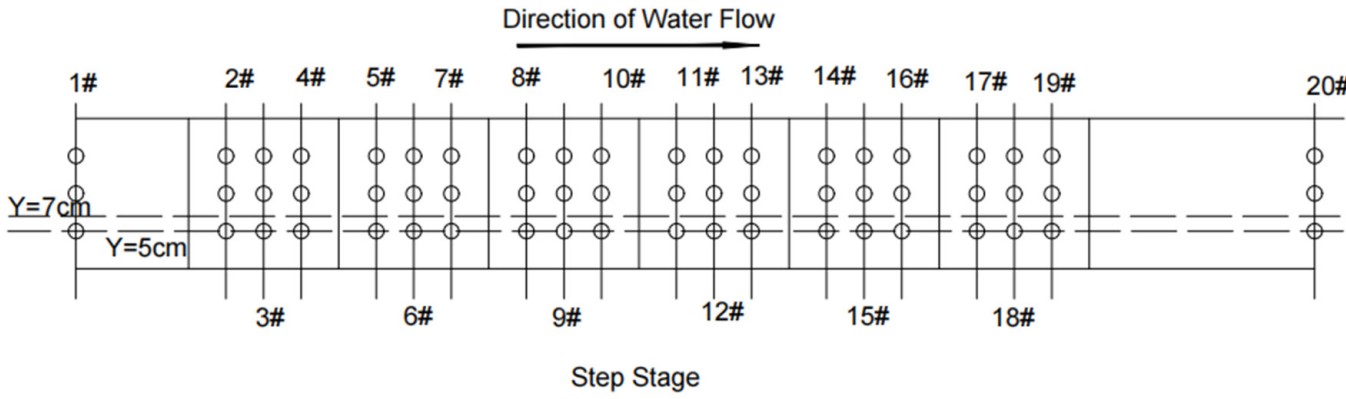

**Figure 5.** Schematic layout of measurement points.

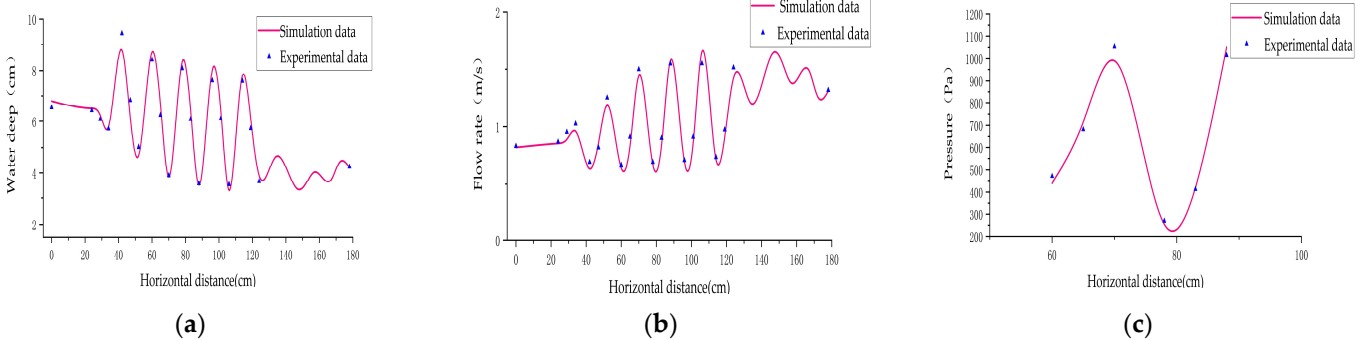

**Figure 6.** Comparison of simulated and measured values of Q = 30 m$^3$/h. (**a**) Water depth along the course, (**b**) flow rate, and (**c**) pressure.

## 3. Results and Discussion

### 3.1. Flow Analysis

Figure 7 shows the water flow pattern of the step energy dissipation structure for different body types at Q = 30 m$^3$/h in the y = 7 cm section. A zero water volume fraction represents air. In the first stage, the water flow is basically the same for steps of all sizes and is relatively stable; the water surface line is wavy, and the water flow starts to mix with gas near the sixth step. In the straight section under the first stage, different degrees of water jumps occur in each body type. When the energy dissipation pier is arranged in the straight section of the step, the point at which the water jumps is shifted backwards to the waterward side of the pier. This accentuates the water jumping effect in the straight section and results in a flow of water more fully mixed with air. The water level also rises after jumping in the staggered arrangement, resulting in better air-doping.

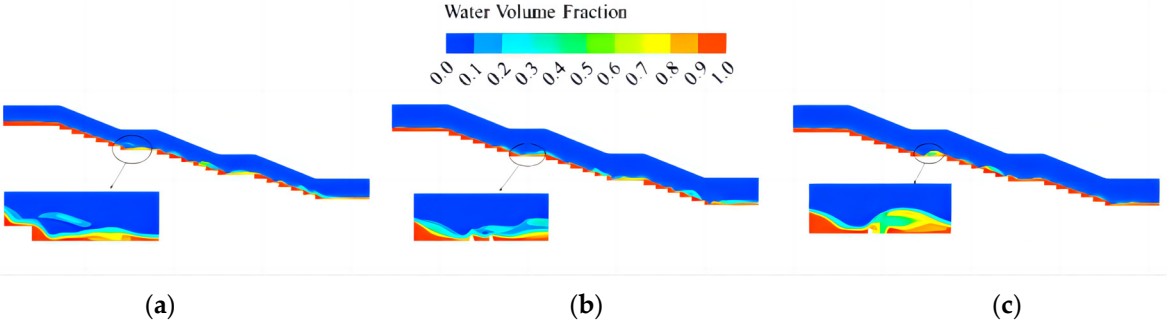

**Figure 7.** Flow pattern of each body type step water flow at Q = 30 m$^3$/h. (**a**) Traditional steps, (**b**) double-row arrangement, and (**c**) staggered arrangement.

In the second stage, the doping concentration of the pier-type dissipator is significantly greater than that of the traditional dissipator. This is due to the fact that the flat section of the first step of the energy dissipation pier increases the doped gas concentration of the water flow so that the water flow is mixed in advance when entering the second stage, thus reducing the length of the non-doped zone and increasing the gas-doping concentration along the second stage. Because the staggered pier arrangement makes the flat section of the water flow more fully doped, this arrangement also results in the highest concentration of doped gas along the second stage. In addition, the water jump occurs in the flat section under the second stage and the third stage so that the water jump effect is enhanced by the use of the energy dissipation pier, and the concentration of gas-doping along the flow is increased. The use of energy dissipation piers to increase the water depth after the jump in the straight section of the step results in an improved gas-doping concentration along the step, a reduction in the length of the non-doped zone, and a lowering of the risk of cavitation damage. The staggered arrangement is more fully dosed than the double-row arrangement and is thus more effective in reducing cavitation.

Figure 8 shows a comparative flow-pattern diagram of the double-row arrangement of the step energy dissipator structure at Q = 30 m³/h, in the longitudinal section, without the energy dissipation pier (y = 5 cm), and with the energy dissipation pier (y = 7 cm). At the end of the flat section, the longitudinal section with the pier (y = 7 cm) produces a large water jump, while the longitudinal section without the pier (y = 5 cm) has no water jump; however, due to the obstructing effect of the pier on the water flow in the flat section, the water jump occurs in the downstream position of the first-stage cyclone, when a large amount of air is mixed in, increasing the air mixing effect of the water flow so that, when the water enters the second stage, no non-air mixing area is produced, effectively reducing the cavitation hazard of the stage; the water flow in the second- and third-stage section exhibits basically the same doping effect, effectively reducing the risk of cavitation damage.

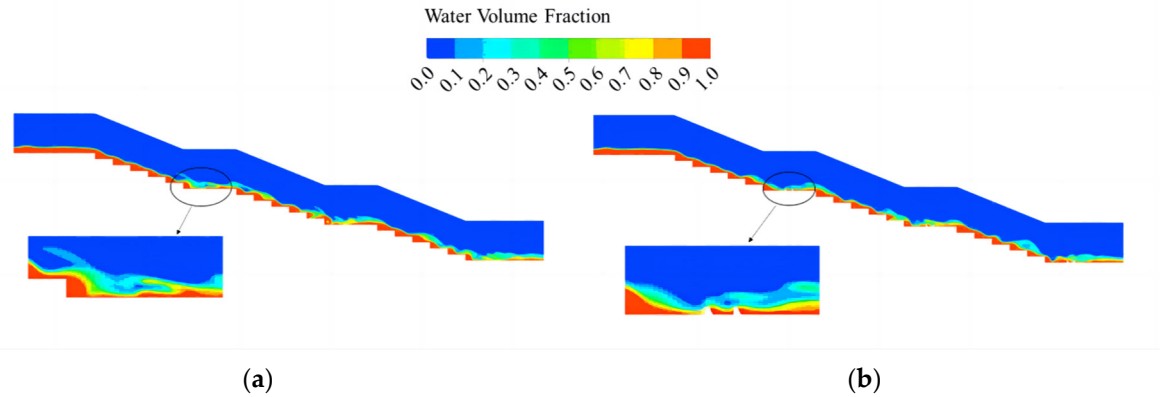

**(a)** **(b)**

**Figure 8.** Flow pattern of different longitudinal sections in a double row arrangement. (**a**) y = 5 cm, and (**b**) y = 7 cm.

Figure 9 shows a staggered pier arrangement of the step energy dissipation structure at Q = 30 m³/h, with two different longitudinal sections, as follows: y = 5 cm for the second row of the energy dissipation pier section, and y = 7 cm for the first row. Water jumps of different degrees can be seen in both sections; those of the y = 7 cm longitudinal section are higher than those of the y = 5 cm longitudinal section, indicating a better gas-doping effect, which is due to the obstruction of the water flow of the first row of energy-dissipating piers so that the water flow is higher when passing through the first row of piers, while the water depth is greater after the second row. Comparing the effects of gas blending in the second and third stages of the two sections, it can be seen that, due to the higher water jump in the y = 7 cm longitudinal section, there is a higher doping concentration along the second stage, because the air-doping escapes along the way and the second straight section does not produce a larger water jump, so the two sections of the third stage of the air-doping effect produce basically the same effect. The staggered arrangement, therefore, enables the entire stage section to be air-doped; however, the air-doping effect does vary from one longitudinal section to another.

*3.2. Flow Rate Analysis*

Figure 10 shows the flow velocity distribution of each step dissipator body type in the y = 7 cm section at Q = 30 m³/h. On the virtual bottom plate formed by the convex angle of the steps, for steps of all sizes, the upper-part flow rate is greater than that of the middle and lower parts, and the highest rates of water flow are mainly seen near the free liquid surface. In the first stage, the flow velocity distribution is basically the same for each type of step dissipator, with an average flow velocity of about 2.0 m/s. In the flat and straight section under the first stage, the segmented pier-type step dissipator forms a water jump behind the pier due to the obstructing effect of the pier on the water flow, resulting in a lower flow velocity in the second stage than the traditional step dissipator. The action of the second stage under the flat section of the energy dissipation pier produces another water jump, resulting in a further reduction in the water flow velocity into the third stage.

The action of multistage energy dissipation piers can reduce the flow velocity to below 1.3 m/s, thus lowering the risk of water scouring damage downstream.

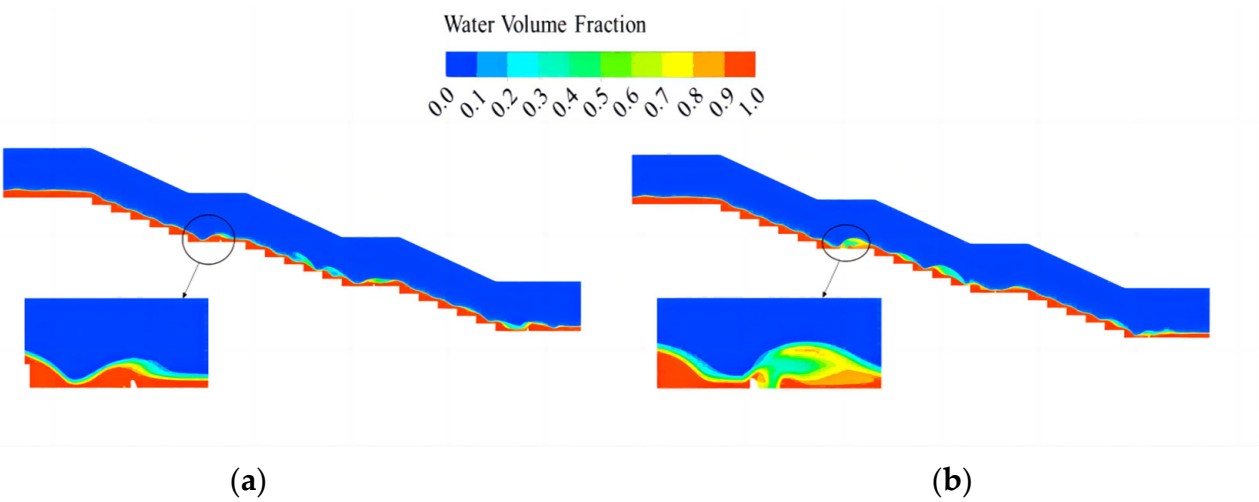

**Figure 9.** Flow pattern of different longitudinal sections in staggered arrangement. (**a**) y = 5 cm, and (**b**) y = 7 cm.

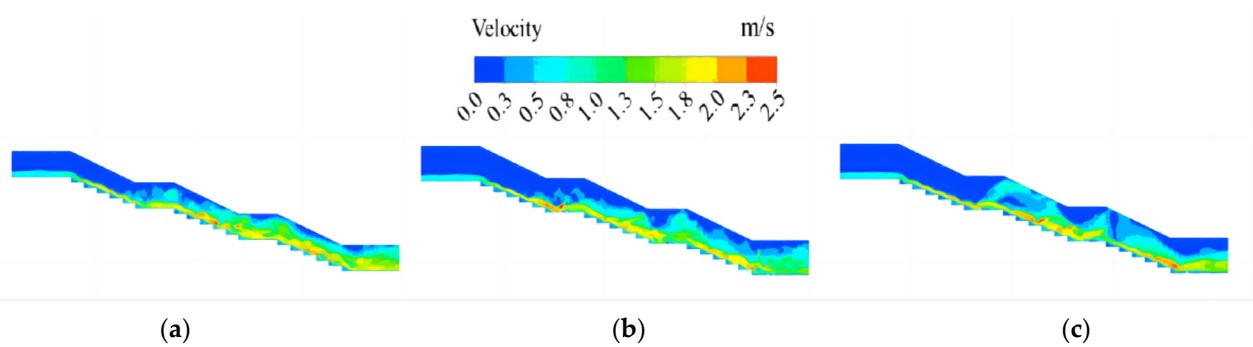

**Figure 10.** Flow velocity diagram of each body type step section at Q = 30 m³/h. (**a**) Traditional steps, (**b**) double-row arrangement, and (**c**) staggered steps.

Figure 10b,c shows that the water surface of the water jump section near the energy dissipation pier produces a large flow velocity of the roll area; the water jump section of the water turbulence intensifies, shear becomes strong, and there is a constant exchange of roll and mainstream, resulting in a large amount of energy loss and reduction in the water flow velocity. Comparing the two forms of the double-row arrangement and staggered arrangement, it can be seen that water flow in the third stage is more stable with a staggered arrangement. This indicates that the staggered arrangement is more effective in improving the water flow structure and stabilising the water flow. The staggered arrangement is also more effective in reducing scour damage downstream, due to the lower flow velocity.

*3.3. Pressure Analysis*

Figure 11 shows the step pressure distribution for the control/traditional arrangement at Q = 30 m³/h. On the horizontal surface of the steps, from the concave angle to the convex angle position, the pressure first increases and then decreases. On each step surface, a higher positive pressure mainly occurs near the convex angle, at about 0.3 times the length of the step position. This is because the water impact on the horizontal surface of the steps at the impact point causes a larger positive pressure; after the impact of the steps, some of the flow continues downstream, while the rest of the water turns to the vertical side of the steps and climbs upwards when it meets them, resulting in a gradual decrease in pressure, which reaches a minimum near the downstream side of the steps. In addition, the climbing

water near the vertical side is guided by the mainstream and, thus, flows downstream, forming a vortex. Negative pressure mainly occurs on the vertical surface of each step, at approximately 0.6–1 times the height of the step. In the straight section of the step, the pressures are all positive. Here, the step water impacting the flat section creates a higher pressure; elsewhere, the pressure distribution is uniform.

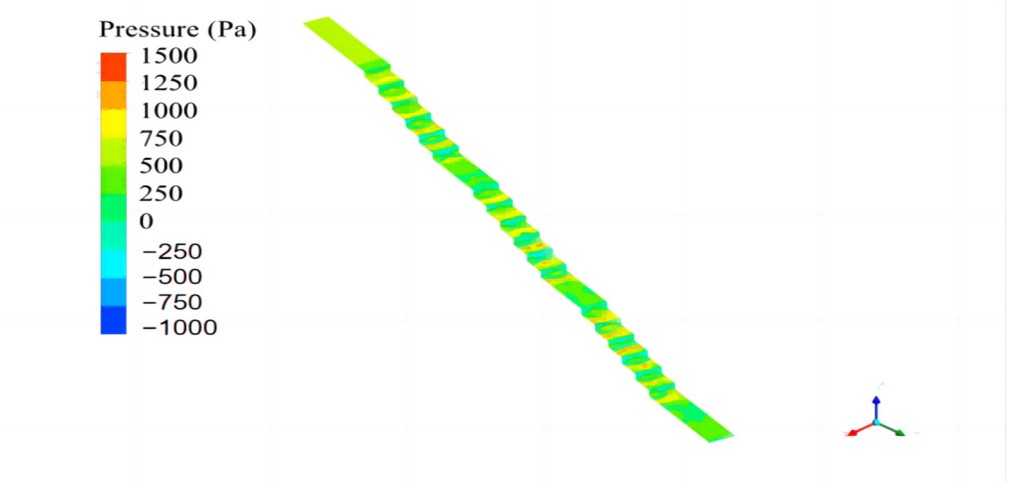

**Figure 11.** Pressure distribution of conventional step dissipator at Q = 30 m$^3$/h.

To study the effect on the pressure of adding energy-dissipating piers in the flat section, we carried out a local analysis of the flat section under the first stage of the segmented plus pier-type step energy dissipator, as shown in Figure 12. The additional energy dissipation piers in the flat section of the steps produce a high positive pressure on the water surface and the upstream area of the piers, and a high negative pressure on the top and rear of the pier, with the negative pressure higher in the first row than the second. Comparing the two arrangements, in the staggered-pier arrangement, the upstream area is larger than the high-pressure area, and the second row of piers is subjected to higher pressure on the water surface. This is due to the staggered arrangement itself, in which each pier is located in only one longitudinal section, maximising the blocking effect of each pier on the higher velocity water flow. This explains why the upstream area is larger than the high-pressure area, and why the second row of energy dissipation piers is subjected to higher pressure on the water surface. However, the negative pressure generated by the staggered arrangement on the top of the piers and on the backwater surface is greater, so cavitation damage is more likely to occur. Corresponding positions in real-life settings should be protected accordingly.

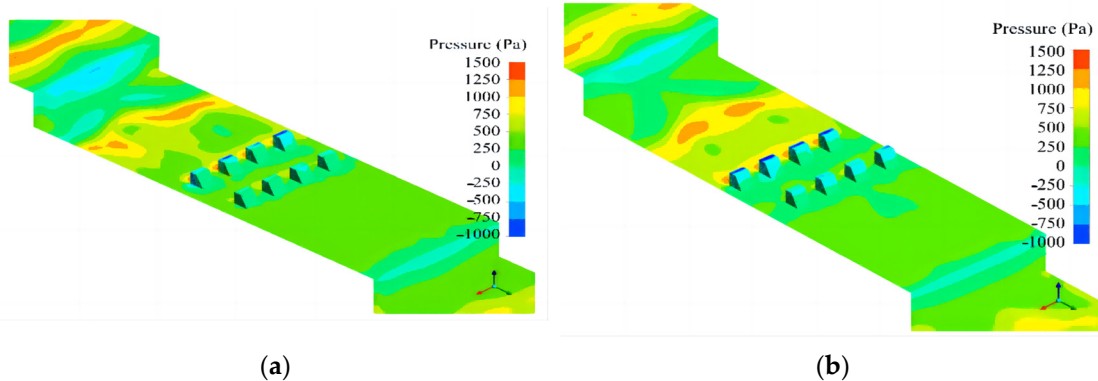

|     |     |
| :-: | :-: |
| (**a**) | (**b**) |

**Figure 12.** Segmented pier-type step dissipation pressure distribution at Q = 30 m$^3$/h. (**a**) Double-row arrangement and (**b**) staggered arrangement.

### 3.4. Energy Dissipation Analysis

In this study, we calculated the energy dissipation rate using the ratio of the difference between the upstream and downstream energy and analysed the step dissipation rates for each body type. Using a *0-0* section of the downstream floor as the reference surface, a *1-1* section at 45 cm from the inlet position as the upstream section, and a *2-2* section at 50 cm from the outlet position as the downstream section, we calculated the energy dissipation rate $\eta$ is as follows:

$$E_1 = \Delta h + \alpha_1 \frac{v_1^2}{2g} \tag{10}$$

$$E_2 = \alpha_2 \frac{v_2^2}{2g} \tag{11}$$

$$\eta = \frac{\Delta E}{E_1} \times 100\% = \frac{E_1 - E_2}{E_1} \times 100\% \tag{12}$$

where $E_1$ and $E_2$ are the total energy of the *1-1* section and the *2-2* section, that is, the upstream and downstream sections; $\Delta h$ is the difference in height between the two sections; $\Delta E$ is the energy difference; $\alpha_1$ and $\alpha_2$ are the flow rate coefficients of the upstream and downstream sections, which generally take 1; $v_1$ and $v_2$ are the average flow rates of upstream and downstream sections; the $\eta$ is the dissipation rate.

Figure 13 shows a comparison of the dissipation rate changes for the energy dissipation structures under each arrangement. The energy dissipation rate of the structure under the step-only section condition exhibits a gradually decreasing trend with increases in the flow. At the time of our test, the lowest energy dissipation rate was 73.44%, and the highest was more than 90%. However, the introduction of the energy dissipation piers changed the local water flow and flow velocity, and produced a high-turbulence kinetic energy dissipation area, with a high-speed consumption of the water body's mechanical energy in the energy dissipation pier at the high-speed consumption, so that the energy-dissipation performance of both the double-row and staggered arrangements is significantly better than the no-pier arrangements, with a maximum difference in the energy dissipation rate approaching 10%. The additional energy dissipation pier in the flat section of the step not only increases the energy dissipation rate but also slows down the reduction in the rate; in this regard, the impacts of the double-row and staggered arrangements are basically the same, and the choice of the arrangement has almost no effect on the energy dissipation rate.

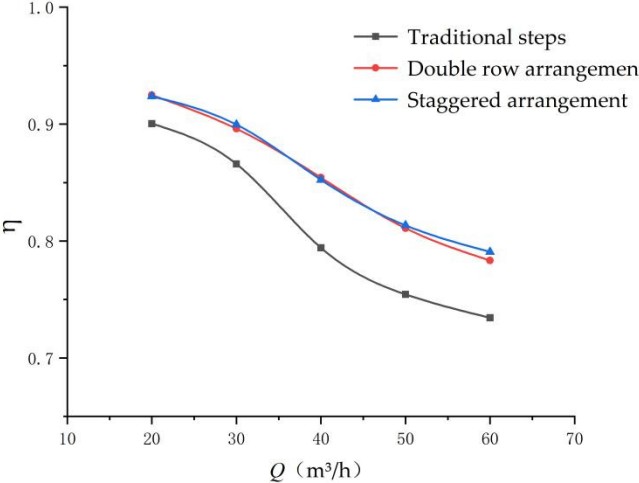

**Figure 13.** Energy dissipation rate of each body type step dissipator.

### 4. Conclusions

(1) The staggered arrangement of energy-dissipating piers causes water to jump back. After the jump back, the water depth increases and this improves the air-doping effect

of the water flow by increasing the air-doping volume, additionally lowering the risk of cavitation damage in this stage. The staggered arrangement results in a better doping concentration than the double-row arrangement; however, the double-row arrangement is better able to reduce the cavitation hazard. Both arrangements exhibit a full section stage-doping, but the staggered arrangement results in different doping concentrations in different longitudinal sections.

(2) The water flow velocity in the segmented energy dissipation pier structure is high at the surface and lower towards the bottom; the water jump generated by the flat section of the pier reduces the flow velocity along the pier and reduces scour damage downstream. The staggered arrangement improves the flow structure and stabilises the flow; the lower flow velocity at the downstream outlet results in reduced scour damage downstream.

(3) The existence of the flat section of the energy dissipation pier in the segmented pier-type structure means that the piers face the water surface, resulting in high positive water pressure upstream, while high negative pressure builds at the top of the pier and the backwater surface. This negative pressure is higher with the staggered arrangement, and this should be borne in mind by engineers involved with real-world structure projects.

(4) Compared with the traditional no-pier design, the segmented pier-added step energy dissipation structure increases the energy dissipation rate by more than 5% on average and produces an improved energy dissipation effect; however, the choice of a double-row or staggered arrangement has little or no effect on the energy dissipation overall.

**Author Contributions:** Data curation, Z.F., Y.L., Y.T. and Q.L.; investigation, Z.F. and Y.L.; writing—original draft, Z.F.; writing—review and editing, Y.L. All authors have read and agreed to the published version of the manuscript.

**Funding:** The research was funded by the National Natural Science Foundation of China (51179116).

**Institutional Review Board Statement:** Not applicable.

**Informed Consent Statement:** Not applicable.

**Data Availability Statement:** Some or all of the data, models, or code that support the findings of this study are available from the corresponding author upon reasonable request.

**Acknowledgments:** This research was supported by the Collaborative Innovation Center of New Technology of Water-Saving and Secure and Efficient Operation of Long-Distance Water Transfer Project at the Taiyuan University of Technology.

**Conflicts of Interest:** The authors declare no conflict of interest. The funders had no role in the design of the study; in the collection, analyses, or interpretation of data; in the writing of the manuscript; or in the decision to publish the results.

## Nomenclature

| | |
|---|---|
| $\rho$ | Density factor derived from weighted average of volume fractions (-) |
| $\mu$ | Molecular viscosity factor derived from a weighted average of volume fractions (-) |
| $G_k$ | Turbulent energy generation term due to mean flow gradient (-) |
| $\mu_t$ | Turbulent viscosity coefficient (-) |
| $\mu_{eff}$ | Effective viscosity (-) |
| $C_{1\varepsilon}, C_{2\varepsilon}$ | Turbulence model coefficients of 1.44 and 1.92, respectively (-) |
| $\alpha_\kappa, \alpha_\varepsilon$ | The inverse of the Planter number for k and $\varepsilon$, taken as 1.0 and 1.3, respectively (-) |
| $C_\mu$ | Experience factor, 0.0845 (-) |
| $\eta_0, \beta$ | Model constants, taking values of 4.38 and 0.015, respectively (-) |
| $Q$ | Flow (m$^3$/h) |
| $E_1$ | 1-1 Total energy of the cross-section (J) |
| $E_2$ | 2-2 Total energy of the cross-section (J) |
| $\Delta E$ | Energy difference (J) |

| $\alpha_1, \alpha_2$ | Flow coefficient at upstream and downstream sections, generally taken as 1 (-) |
| $v_1$ | Average cross-sectional flow rate upstream (m/s) |
| $v_2$ | Average cross-sectional flow rate downstream (m/s) |
| $\eta$ | Energy dissipation rate (-) |

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
