# Peer review of "Effects of Energy Dissipation Pier Arrangements on the Hydraulic Characteristics of Segmented Pier-Type Step Energy Dissipator Structures"

_water, doi:10.3390/w14223590_

Round 1

Reviewer 1 Report

An interesting topic has been chosen, but the writing of the results is weak. I think that the paper can be sensibly improved. In particular, please consider the following recommendations:

1-      There are several language/grammatical problems within the manuscript, please get help from a native editor.

2-      The title is long and I suggest shortening it.

3-      The manuscript section should be adjusted according to the Instructions for Authors. Research manuscript sections: Introduction, Materials and Methods, Results, Discussion, Conclusions (optional).

4-      In the last paragraph of the first part, sentence "Therefore, in this paper, we propose a segmented pier ….." should be written as a Passive form.

5-      In choosing the optimal turbulence model, has the performance of other models been checked? The result of checking the performance of the models presented in a table.

6-      After Equation 9, the parameters are given in italics.

7-      It is suggested to use numerical model instead of mathematical model.

8-      It is better to present the studied parameters and their range of changes in a table.

9-      Figure 2 is unclear.

10-   According to Figure 3, it is possible that the relative error percentage will decrease by reducing the mesh size. It is better to present the mesh size less than 0.006 m in the diagram. It is also suggested to choose the vertical axis of relative error percentage.(Please see: https://doi.org/10.1016/j.flowmeasinst.2020.101810)

11-   Figure 5 is unclear.

12-   In Figure 6, the titles of the horizontal axes are not selected correctly. It needs to be checked.

13-   The quality of Figure 6 should be increased.

14-   The quality of Figure 7 is awful. Improve its quality and clarity.

15-   The results should be reported quantitatively and with tables or graphs. The results section needs serious revision.

16-   Authors should provide diagrams that clearly show the research objective.

17-   Figures 10, 11, 12 and 13 also need improvement.

18-   Sections are undefined and shown on the figure.

Reviewer 2 Report

This paper proposes an innovative staggered arrangement of energy dissipation piers which can increase the gas doping effect of water flow and reduce the cavitation damage of the stage. The staggered arrangement can also improve the water flow structure and stabilize the water flow, with lower flow velocity at downstream outlet, thus reducing the scouring damage downstream. The findings are well supported by the analysis results provided. The paper is well written and is of readership to the community. I recommend acceptance of this paper.

Reviewer 3 Report

The manuscript titled "Study on the Effect of Energy Dissipation Pier Arrangement on the Hydraulic Characteristics of Segmental Additional Pier Type Step Energy Dissipator" is worth exploring. The authors propose a segmented pier type step energy dissipation worker, and adopt numerical simulation method to study the influence of the arrangement of energy dissipation pier on the hydraulic characteristics of the segmented pier type step energy dissipation worker, and explore the best arrangement of energy dissipation pier, so as to provide some reference basis for improving the air mixing effect of the step water flow, increasing the energy dissipation rate and the related structure design. The article is well written but some points should be considered and revised.

Since line numbers are missing, I will leave only general comments.

Abstract and conclusion are very general. For this work the most important data should be described in Abstract and Conclusion to make the article more attractive for readers.

Some figures (2, 5, 12) are not readable. As I understand it is not the authors falls, but feature of PDF structure made by the system. Nevertheless, there is no opportunity to evaluate the listed figures.

References should be reached by newer sources to point out relevance of the topic. Or topic is not relevant at this moment?

Perhaps conventional structure of the article including Introduction, M and M, Results, Discussion and Conclusion sections will be better perceived by readers

Scientific soundness of the work is low, but it can be interesting for industrial readers and engineers. In my opinion the manuscript should be revaluated after revision.

Round 2

Reviewer 1 Report

Dear Editor,

Thank you very much for your invitation to review the revised manuscript.
The revised manuscript is improved and the comments of reviewers were sufficiently responded and/or addressed.
Regards,

Reviewer 3 Report

The authors considered all my comments and recommendation and decided them mainly well. Conclusion still can be improved and reached be more data. My comment regarding scientific soundness and rates of originality/novelty, significance of content and quality of presentation stay the same. Readers' interest in the revised article will be much higher than in the previous version. Therefore, at this stage I would recommend revised version of the article “Effects of Energy Dissipation Pier Arrangements on the Hydraulic Characteristics of Segmented Pier-type Step Energy Dissipator Structures” for publication in Water.